# The Origin Recognition Complex: From Origin Selection to Replication Licensing in Yeast and Humans

**DOI:** 10.3390/biology13010013

**Published:** 2023-12-25

**Authors:** Bik-Kwoon Tye, Yuanliang Zhai

**Affiliations:** 1Department of Molecular Biology and Genetics, Cornell University, Ithaca, NY 14853, USA; 2School of Biological Sciences, The University of Hong Kong, Hong Kong, China; zhai@hku.hk

**Keywords:** origin selection, replication initiation, eukaryotic DNA replication, cryoEM structures, MCM hexamers, origin recognition complex, Meier–Gorlin syndrome

## Abstract

**Simple Summary:**

The origin recognition complex (ORC) selects sites for replication initiation by recruiting a pair of hexameric minichromosome maintenance (MCM) complexes to replication origins where the pre-replication complex (Pre-RC) is assembled, and the bidirectional replisomes are formed. In yeast, site selection by ORC is largely based on sequence-specific binding and Pre-RCs are established where ORC binds. In humans, site selection by ORC is largely based on chromatin landscapes and Pre-RCs are formed far away from ORC binding sites. This review compares these two very different modes of origin selection that can be traced to a small variation in the structures of subunit 4 of the yeast and human ORC.

**Abstract:**

Understanding human DNA replication through the study of yeast has been an extremely fruitful journey. The minichromosome maintenance (MCM) 2–7 genes that encode the catalytic core of the eukaryotic replisome were initially identified through forward yeast genetics. The origin recognition complexes (ORC) that load the MCM hexamers at replication origins were purified from yeast extracts. We have reached an age where high-resolution cryoEM structures of yeast and human replication complexes can be compared side-by-side. Their similarities and differences are converging as alternative strategies that may deviate in detail but are shared by both species.

## 1. Introduction

Eukaryotic DNA replication is highly conserved with one guiding principle: the entire genome must be replicated precisely and accurately once in every cell division. It is not surprising that a fundamental mechanism shared by all eukaryotes is highly conserved, but how conserved is it? The replication of DNA must be coordinated with the gene expression program of a cell to synchronize with the temporal expression of genes and to limit conflicts of DNA replication and transcription forks, which are a major source of DNA damage. Yeast is an excellent model for studying DNA replication because of its simplicity, but its life cycle is very different from those of vertebrates that have elaborate developmental programs. In this review, we will use the study of the origin recognition complex (ORC) as an example for how one can learn about human DNA replication through the study of yeast, with an emphasis on studies from our recent work [1,2,3,4]. For a more comprehensive review on the influence of chromatin on the timing and site selection by ORC, we refer you to a recent review by Lee et al. [5].

## 2. Replication Licensing in Cell Cycle and Developmental Regulation

The assembly process of the replisome at a replication origin has been recapitulated in vitro using a purified yeast system [6,7,8,9,10,11,12]. In the early G1 phase of the cell cycle, the ORC binds a replication origin and with the help of Cdc6, recruits the MCM-Cdt1 heptamer, one at a time, to form a head-to-head MCM double hexamer (MCM-DH) [13,14,15,16]. The MCM-DH forms an inert ring structure around the origin DNA upon loading by ORC in a step known as replication licensing. This licensed pre-replication complex (Pre-RC) is activated through the concerted effort of two kinases, the Dbf4-dependent kinase (DDK) [4,17,18,19,20] and the cyclin dependent kinase (CDK), in addition to several firing factors as the cell enters the S phase [6,13,20,21,22]. The activated pre-RC then serves as the scaffold for the sequential assembly of the bi-directional replisomes. Inactivation of ORC and MCMs by CDK phosphorylation prevents pre-RC reassembly from late G1 to mitotic exit [23,24,25]. More recently, the human replisome has also been reconstituted using purified proteins in an assembly process largely similar to that of the yeast replisome [26].

Due to the very large size of the eukaryotic genome, it must be replicated in segments referred to as replicons, each with its own initiation site, to complete DNA synthesis within the restricted timeframe of the S phase. So, the length of the S phase is closely correlated with the replicon size and the actual number of initiation sites [27]. It is known that the rate of cell division is limited by the length of the S phase in rapidly dividing cells [28,29]. If ORC plays a universal role in the selection of replication origins, its function may help to explain the phenotype of the Meier–Gorlin Syndrome (MGS), a primordial dwarfism in humans. MGS is an inherited condition traced to autosomal recessive mutations mapped in genes involved in DNA replication initiation such as ORC1, ORC4, ORC6 and CDC6 [30,31,32]. MGS patients are proportionate dwarfs that have a miniature stature and organs as if a reduced number of cell divisions occurred at every critical juncture of development such that all organs and tissues are formed but in greatly reduced proportions. If origin choice is influenced by ORC, then ORC potentially plays an important role in the once per cell cycle regulation of DNA replication, the cell division rate, as well as the overall development of complex organisms in their life cycles. This line of reasoning for ORC involvement may also apply to cancer development and the plasticity of origin selection in transformed cells.

## 3. The Origin Recognition Complex Is Conserved in Structure but Diverged in DNA Binding Properties

The origin recognition complex is a six-subunit complex that was initially purified in yeast based on its affinity to yeast origin DNA in an ATP-dependent manner [33]. Yeast replication origins can be cloned on plasmids as autonomously replicating sequences (ARSs) [34,35]. These ARSs share a 17 bp consensus sequence known as the ARS consensus sequence (ACS) [36,37]. The six ORC subunits, Orc1, Orc2, Orc3, Orc4, Orc5 and Orc6, are found in all eukaryotes. They are highly conserved in protein sequence and structure from yeast to humans. However, despite their conservation in structure, ORC has diverged binding sites ranging from an absolute requirement for a specific sequence in the budding yeast to a total agnosticism to base sequences in humans. The binding of the human ORC on chromatin appears to be dictated by the chromatin landscape, clustering in intergenic regions [38] where nucleosomes are depleted, such as CpG islands, transcriptional start sites and GC-rich DNA [39]. An immediate question is why do metazoans and fungi use different strategies for identifying replication initiation sites? Humans developing from a single fertilized egg to an adult go through many stages of development that require the programmed expression of different sets of genes. As a result, the chromatin landscape changes throughout development. Just like transcription factors, replication initiators need to find their target sites in response to these changes [40,41]. On the other hand, yeasts are single-celled organisms that have hardly any developmental changes. Their life mission is to divide as rapidly and efficiently as possible. Their chromatin landscape is largely constant.

Like metazoan replication origins, the budding yeast ARSs are located at intergenic regions. They are nested in nucleosome free regions of about 125 bp in polar T-rich and A-rich DNA segments. The essential element, ACS, is a 11 bp sequence (WTTTATRTTTW) [42] asymmetrically positioned near the upstream nucleosome while two important elements, B1 and B2, are located downstream from the ACS [37]. Upon ORC binding to the ACS, the downstream nucleosome is repositioned [43] presumably to make room for the loading of the MCM double hexamer at the B2 element. Of the more than 400 ARSs identified, 249 have these canonical features [44].

## 4. The cryoEM Structure of Yeast ORC Bound to Origin DNA

To understand the interactions of the yeast ORC with origin DNA, it is important to study its structure at atomic resolution. Yeast ORC (yORC) bound to origin DNA was reconstituted and subjected to single-particle cryo-electron microscopy analysis [2]. The structure of the yORC bound to 72 bp of ARS305 DNA containing the ACS and B1 element was determined to 3.0-Å resolution showing five of the subunits, Orc1-5, encircling the ACS (Figure 1a) while Orc6 is situated away from the ACS (Figure 1b). A prominent feature of this ORC-ARS DNA complex is the bending of the origin DNA at successive points at the ACS and B1 element (Figure 1c). When this structure is superposed with the structure of the ORC-Cdc6-Cdt1-Mcm2-7 (OCCM), an intermediate captured in the presence of ATPγS during helicase loading [45], the bent DNA aligns with the Mcm2-Mcm5 gate [2]. The implication of the orientation of the bent DNA with respect to the Mcm2-Mcm5 gate is that DNA bending facilitates a precise insertion of origin DNA into the MCM ring during the formation of the OCCM intermediate. Indeed, bending of the origin DNA is also observed in Drosophila ORC [46]. The salient features of yORC complexed with ARS305 that help explain the properties and function of ORC are described below.

## 5. Yeast Orc4 Insertion Helix Encodes Sequence-Specific Binding

The yORC binds origin DNA in two modes: base-specific and base non-specific. Orc1-5 each interacts with the DNA backbone along the ACS to the B1 region at multiple points, creating a tight grip. This non-specific DNA binding property explains the affinity of ORC even for non-ARS DNA. In addition to the base non-specific binding, three of the ORC subunits interact specifically with the bases of the ACS. A basic patch (BP) from Orc1 and the Orc2 initiator-specific motif (ISM) interact with the minor groove of the ACS while an insertion alpha helix (IH) in Orc4 interacts with the major groove (Figure 1c). The hydrophobic residues of the IH are embedded in a hydrophobic environment created by the methyl groups of the invariant Ts of the ACS, forming strong interactions. An alignment of the beta-hairpin region containing the Orc4 IH showed that the IH sequence has diverged among yeast species that have different ACSs but are totally absent in metazoans (Figure 1d). For the fission yeast S. pombe, ORC binds AT-rich sequences and the IH is missing, instead SpOrc4 has acquired four A-T hooks [47,48]. As the Orc1-BP and Orc2-ISM interact with the minor groove, where A-T or T-A base pairs are indistinguishable [49,50], of the three DNA binding motifs, the IH of Orc4 is the most likely determinant for the sequence-specific binding property of *Saccharomyces cerevisiae* ORC (ScORC). If so, removal of the IH of Orc4 should eliminate the sequence-specific binding property of the *Sc*ORC and convert it to one with properties more akin to metazoans such as Drosophila and humans (Figure 1e–j).

## 6. Humanizing the Yeast ORC by Removing the 19 Amino Acid Insertion Helix of Orc4

To test this hypothesis, a mutant yeast strain (*orc4-IH*D was constructed such that the 19 amino acid insertion helix was removed from Orc4 [3]. Unexpectedly, this strain was viable albeit showing a prolonged G2 phase and an activated checkpoint in the late S phase as indicated by the phosphorylation of Rad53. Collisions between replication and transcription forks are a likely source of fork stalling and checkpoint activation in an uncoordinated replication initiation program. Footprints of ORC based on ChIP-seq showed that the mutant ORC (referred to as ORC-IHD) now binds many more sites in clusters throughout the yeast genome with little overlap with the canonical ARSs, indicating that ORC-binding shows an apparent decrease in sequence specificity (Figure 2a–c). As a result, ORC-IHD favors binding to transcriptional start sites (TSS) with 83% of binding sites located within 500 bp upstream of a TSS. Furthermore, this binding positively correlates with the transcriptional strength of the promoter (Figure 2d). In fact, these characteristics are precisely the binding profile observed for human ORC [39,51,52]. TSSs are known to have a polar bias in T-rich and A-rich sequences and well-positioned downstream nucleosomes, as found at ARSs in *S. cerevisiae*, suggesting that the mutant ORC-IHD has retained many of the site-selection properties but lost its base sequence specificity.

A comparison of the upstream and downstream nucleosome positions relative to the ORC-binding sites in wild-type and mutant cells showed that the mutant ORC-IHD binds extra-wide nucleosome depleted regions that do not require the repositioning of flanking nucleosomes for MCM loading. These observations suggest that the site selectivity of the mutant ORC is now predominantly determined by the chromatin context rather than by base sequence specificity and that ORC-IHD may have lost its ability to reposition the downstream nucleosome (Figure 3). Clearly, the viability of the mutant yeast *orc4-IH*D strain suggests that sequence-specific replication initiation is important but not essential for life in yeast.

Genome-wide analysis of the function of ORC-IHD in MCM loading and replication showed a very different pattern between wild-type and mutant ORC. ORC binding sites can be divided into three categories in these strains: ORC binding sites that are unique to the WT strain, ORC-IHD binding sites that are unique to the mutant strain, and ORC binding sites that are overlapping in both strains (Figure 4). Heatmaps of ORC-, MCM-ChIP seq and BrdU-IP seq tell a very intriguing story. At ORC peaks that are unique to the WT strain, the ORC peaks and MCM peaks are coincident while some ORC-IHD binding near these sites are observed with commensurate loading of MCM and DNA replication activities. At ORC peaks common to both WT and mutant strains, ORC loading of MCM are evident at these sites as well as the corresponding DNA replication activities. Intriguingly, at ORC peaks unique to the *orc4-IH*D mutant, there is no apparent MCM loading at these sites nor DNA replication activities even in the mutant strain. However, in vitro assays showed that Pre-RCs were assembled with comparable efficiencies by mutant and WT ORC. The genome-wide analysis suggests that unlike the WT ORC, which loads and assembles the MCM double hexamer on-site, the locations for MCM loading and MCM-DH assembly were not the same for the humanized ORC. Does the humanized yeast model reflect how human ORCs load and assemble the MCM double hexamer? Some insight into this seemingly inexplicable result is provided in a recent finding that the human Pre-RC (hPre-RC) forms an open complex in contrast to the closed complex formed by the yeast Pre-RC (yPre-RC). We will discuss this contrasted observation in a later section. 

## 7. Interdependence of ORC and the Nucleosomal Context in Origin Site Selection

The budding yeast also uses other mechanisms besides base sequence recognition to recruit ORC to specific sites. It is known that the BAH domain of Orc1, which interacts with the histone H4 tail, is another means of recruiting ORC to ARSs [53]. However, another study shows that the transcription landscape directs replication initiation by the binding of Orc1 to the 5′ ends of transcripts [54]. A recent study showed that Orc1 mediates the nucleosome re-organization by chromatin remodelers such as INO80, ISW1a, ISW2 and Chd1, suggesting that ORC also plays an active role in chromatin reorganization [55]. 

## 8. The Importance of Site-Specific Replication Initiation in Yeast and in Humans

Is ORC-directed site-specific initiation also important in humans? The answer is yes, but humans use different site-specific selection mechanisms that achieve the same goals. As the chromatin landscape changes from embryogenesis to tissue differentiation, site-specific initiation by post-translational modification (PTM) of histone H4 most likely plays a role. The BAH domain of hORC1, where the Meier–Gorlin syndrome mutations map, specifically binds H4K20me2 [31]. Indeed, dimethylation of histone H4K20 is a mechanism for recruiting ORC. The histone H2A.Z, which recruits the histone lysine N-methyltransferase SUV42OH1 for the dimethylation of H4K20 at CpG island promoters, helps to recruit ORC [56] as observed in their colocalization at the borders of high-density hMCM-DH clusters [4]. A classic example of PTM of histones defining sites of replication initiation is the human beta globin gene, which has a single origin for early firing in erythroid cells. The histone acetyltransferase binding to ORC1 (HBO1) targets this origin by acetylating histone H4 [57]. So, it appears that humans and yeast use very similar strategies but widely different mechanisms to manage the replication of their genomes to suit their distinct lifestyles and life cycles. 

## 9. Pre-RC Assembly in Yeast and in Humans

While site-specific initiation is used in a minority of the replication initiation events in humans, it is a predominant mode of replication initiation in yeast. Conversely, while the chromatin landscape is a determining factor in the initiation of DNA replication in humans, yeasts also have the capacity to use landscapes to guide replication initiation. The ORC’s role in both modes of initiation is to facilitate the assembly of the MCM double hexamer at the directed sites. How does it do it?

For yeast, a mechanism for a single ORC to recruit two head-to-head MCM hexamers at canonical ARSs that have two potential binding sites (the ACS and B2 elements) in inverted orientations has been proposed [15,16,58]. In this scenario, a single ORC binds ACS and recruits the first Cdt1-MCM heptamer before performing an acrobatic flip to bind the B2 site and recruit the second Cdt1-MCM while Orc6 is still attached to NTD of Mcm6 in the first MCM hexamer (Figure 5) [15,16]. Alternatively, two ORCs, one binding to the ACS and another at the B2 element, can achieve the same feat in establishing a head-to-head MCM double hexamer [58]. In these inverted ORC-binding sites in proximity, a MCM double hexamer is recruited at or very near to the site where ORC is bound, and replication initiation takes place. This mechanism is unlikely to work in the case of humans where replication origins are devoid of conserved sequence motifs in proximity specific for ORC binding [59,60]. Without a second binding site in proximity, wherever ORC binds, it will only recruit a single MCM hexamer [61]. When ORC binds in clusters, clusters of MCM hexamers will be loaded and head-to-head double hexamers will be formed when two head-to-head hexamers collide after traveling a certain distance [58]. The two different ways of producing the final product of the head-to-head MCM-DH may help explain the two very different MCM-DH or Pre-RC structures formed in yeast and in humans. In wild-type yeast, the MCM double hexamer is formed at locations close to the ORC binding site while in the humanized ORC, MCM locations are non-coincident with ORC binding sites (Figure 4) [3]. These results suggest that in yeast, when replication initiation does not involve the ACS and B2 elements, MCM-DHs may be formed at locations away from ORC by traveling from the MCM loading sites as shown in the case of the humanized ORC. 

This observation is further confounded by a recent study showing that the human pre-RC is an open complex with an initial open structure (IOS) at the interface of the hMCM-DH and that this IOS is essential for the establishment/maintenance of the hPre-RC (Figure 6) [4]. The tight junction formed at the IOS suggests that a major conformational change occurs when the two MCM hexamers collide followed by untwisting of the bound DNA (Figure 5). The structures of yPre-RC, which are reconstituted on engineered canonical ARS DNA, is a closed complex with the encapsulated DNA in the MCM-DH central channel forming a perfect B-form duplex (Figure 6) [62,63]. However, the DNA within the high-resolution structure of the MCM-DH purified endogenously from yeast cells could not be resolved [1]. The technical failure was likely due to the loss of the mobile endogenous genomic DNA bound in the central channel of the yPre-RC [61,64,65]. Therefore, one cannot rule out the possibility that Pre-RC also forms open complexes in yeast at a minority of initiation sites where two MCM hexamers collide after travelling from distant loading sites. Examination of the structure of the Pre-RCs formed in the yeast *orc4-IH*D strain containing the humanized ORC may prove interesting.

## 10. Perspective

The study of eukaryotic DNA replication has benefited tremendously from the study of yeast as a model. Not surprisingly, this highly conserved mechanism shared by all eukaryotes has very similar machinery as shown in the structures of ORC from yeast and humans, but the details of their site selection mechanisms vary to suit their unique lifestyles and life cycles. The base-sequence-specific determinant of the *S. cerevisiae* ORC is encoded by the insertion helix of subunit Orc4, a feature that is missing in metazoans and *S. pombe* (Figure 1). However, humans use other means, such as posttranslational modifications of histone H4 tails [56,57], to recruit hORC to the designated locations. Perhaps, as shown in yeast, chromatin remodelers may be recruited for the reorganization of local chromatin for MCM loading [55]. Chromatin remodeling at canonical ARSs in yeast may have acquired this built-in feature customized for the yORC to bind and reorganize the downstream nucleosome. The different origin selection mechanisms may have contributed to the very different structures of the Pre-RCs in yeast and in humans. High-resolution structures showed that the yeast Pre-RC is a closed complex located at the site of loading by ORC at designated locations, where replication initiations are programmed to suit the lifestyle and life cycle of yeast. In contrast, in humans where the chromatin landscape changes with the expression program of each developmental stage, hORC binds stochastically to open regions of chromatin to load single MCM hexamers [61]. These single hexamers must migrate along DNA, sometimes dodging nucleosome obstacles to encounter another oncoming MCM [66]. These sites of encounter are the potential sites of replication initiation where hPre-RCs are formed. They are marked by the tight junctions of zinc fingers entwined at the NTDs of the respective MCM hexamers (Figure 6). The tight grip of the zinc fingers of hMcm2 and hMcm5 on DNA at the interface followed by a rotational conformational change most likely gives rise to the initial open structures (IOSs) that render the hPre-RC immobile [4]. In this way, human replication origins are marked not at the ORC loading site but at the pre-RC assembly site and that clustering of MCM loading in zones improves the probability of pre-RC formation by head-to-head collisions of the sliding MCM hexamers within individual zones.

Through the study of yeast, much has been learned about human DNA replication and in every case, the similarities are striking. Features unique to yeast or humans that seemed to divide the two systems often emerge as alternatives overlooked in the other system. Indeed, the importance of yeast as a model system in the study of DNA replication cannot be overstated. The increasing number of emerging high-resolution structures of human DNA replication complexes derived from a strong foundation of yeast studies sets the stage for future drug development targeted at human diseases related to DNA replication defects.

## Figures and Tables

**Figure 1 biology-13-00013-f001:**
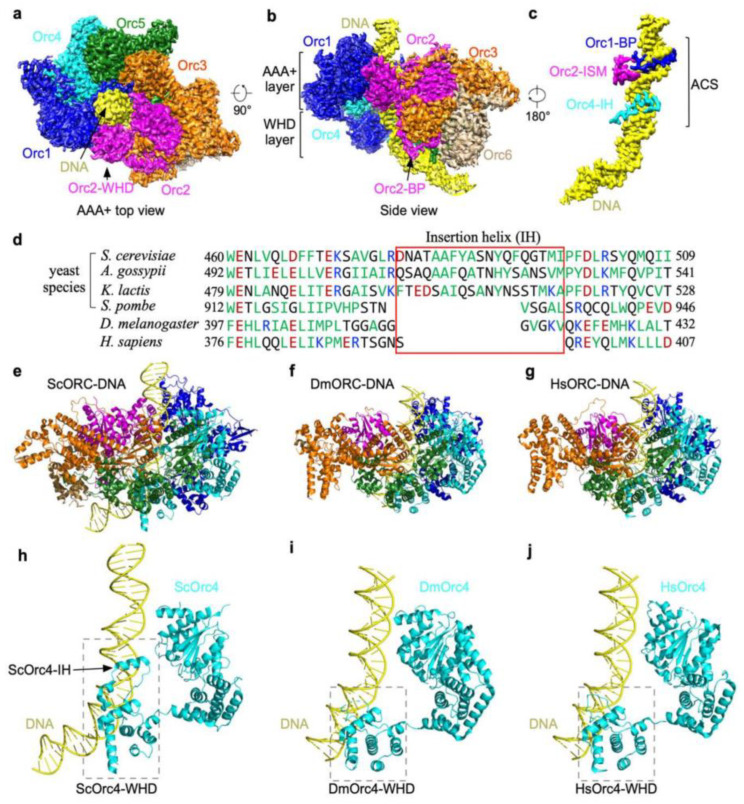
Base-specific recognition of ACS by yORC. (**a**). Top view. (**b**). Side view of ORC bound to ARS305. (**c**). Orc1 basic patch (BP) and Orc2 initiator-specific motif (ISM) interact with the minor groove while Orc4 insertion helix (IH) interacts with the major groove. (**d**). Multiple protein sequence alignment of Orc4 insertion helices from indicated species. (**e**–**g**). The atomic models of the ORC-DNA structures from ScORC-DNA (PDB: 5ZR1) (**e**). DmORC-DNA (PDB: 7JK5) (**f**). and HsORC-DNA. (**g**). Note that the DNA from the DmORC-DNA was modeled into the HsORC structure (PDB: 7JPS). (**h**). ScOrc4-IH interacts with the major groove of ACS. (**i**). IH absent in DmOrc4-WHD. (**j**). IH absent in HsOrc4-WHD.

**Figure 2 biology-13-00013-f002:**
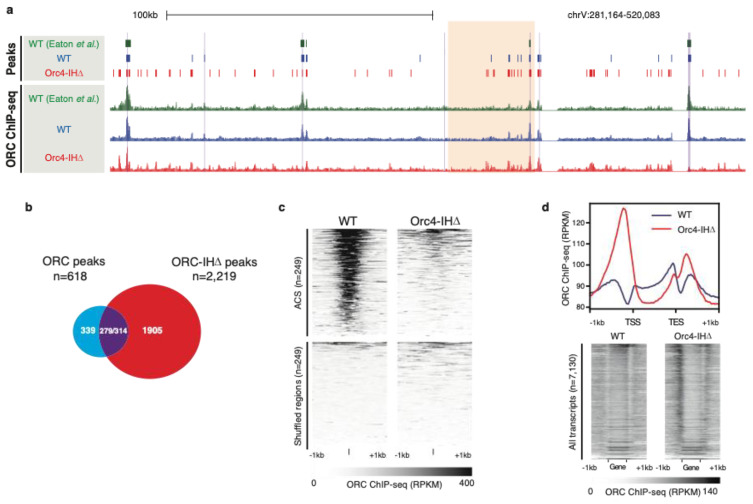
(**a**). Genome browser screenshots illustrate the enrichment patterns of ORC across a region on chrV. Normalized signals (Reads per kilobase per million reads (RPKM)) of WT ORC (blue), ORC-IHD (red), and previously published ORC-ChIP-seq datasets (green). (**b**). Venn diagram showing the overlap between ORC ChIP-seq peaks defined in WT (*n* = 618; blue) and Orc4-IHD cells (*n* = 2219; red). Of these, 279 ORC and 314 ORC-IHD peaks overlapped by at least 1 bp (purple), while 339 and 1905 peaks were identified as unique in WT and Orc4-IHD, respectively. (**c**). Heatmaps show the ORC enrichment patterns in WT and *orc4-IH*D at previously defined ACS (±1 kb of ACS; top). Size- and number-matched shuffled genomic loci were included as controls (bottom). (**d**). Line plot and heatmaps show the ORC enrichment patterns in WT and *orc4-IHD* at annotated transcripts and flanking sequences. The line plot (top) shows the aggregated ORC ChIP-seq signal (RPKM) at all annotated transcripts and the surrounding regions (±1 kb). ORC4-IHD binding (red) at the 5′ of the transcriptional start site (TSS) is dramatically higher than that of the WT ORC (blue). Heatmaps demonstrate the differential enrichment of ORC4-IHD upstream of the TSS. The transcripts in the heatmaps are arranged by descending RNA-seq signals, indicating the association of mutant ORC binding at these loci and the expression levels. Figure reproduced from [3].

**Figure 3 biology-13-00013-f003:**
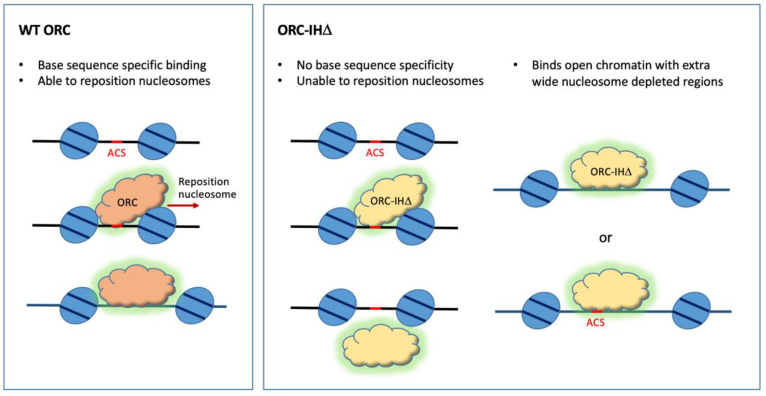
Model for the selection of binding sites by the WT and mutant ORC. WT ORC binds ACS and can reposition the flanking nucleosomes upon binding. ORC-IHD binds promiscuously to T-rich sequences, including ACSs, but is unable to reposition nucleosomes. Therefore, it binds wider nucleosome-depleted regions that do not require the repositioning of nucleosomes for MCM loading.

**Figure 4 biology-13-00013-f004:**
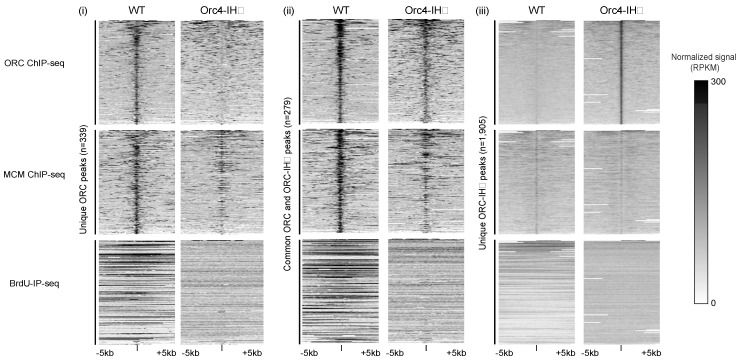
Heatmaps demonstrate the global changes in ORC and MCM ChIPseq and BrdUIP-seq signals. Focusing on ±5 kb surrounding ORC ChIP-seq peaks unique to WT (**i**), common to both WT and ORC4-IHD (**ii**), and unique to ORC4-IHD (**iii**), the normalized signals (RPKM) in WT and *orc4-IH*D cells are shown. Figure reproduced from [3].

**Figure 5 biology-13-00013-f005:**
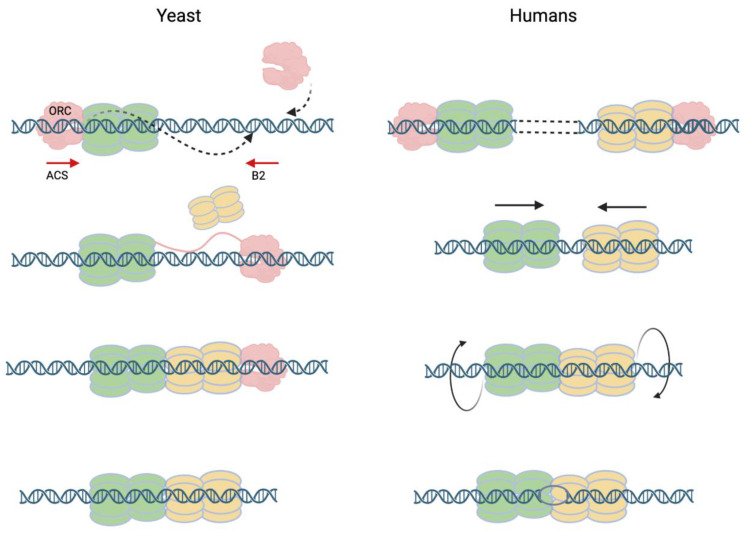
Models for pre-RC assembly after MCM loading by ORC in yeast and in humans. In yeast, canonical ARSs containing ACS and B2 elements provide two binding sites for ORC in inverted orientations. A single ORC binds ACS to recruit the first Ctd1-MCM heptamer (green) then flips to bind the B2 element while still attached to the first MCM and recruits the second Ctd1-MCM heptamer (yellow), allowing the NTDs of the two MCM hexamers to fuse [15,16]. Alternatively, two ORCs, each binding to ACS or B2 simultaneously to recruit two MCM hexamers independently in head-to-head orientations, allow the formation of a head-to-head double hexamer. In humans, ORCs spaced far apart recruit single MCM hexamers located at a distance. The migration of single hexamers in head-on collisions creates MCM double hexamers with tight junctions [61]. Rotation of MCM hexamers in opposite directions results in the formation of an IOS at the hexamer interface ^6^ that renders the hPre-RC immobile.

**Figure 6 biology-13-00013-f006:**
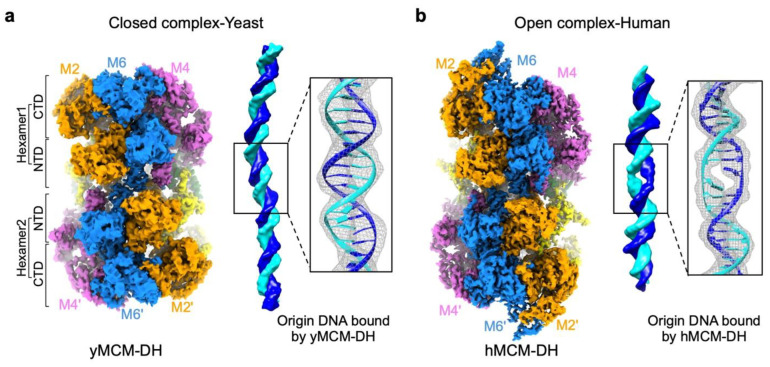
Structures of pre-RCs from (**a**). yeast (EMD-9400, PDB-5BK4) and (**b**). humans (EMD-32258, PDB-7W1Y) have similar overall structures but captured DNA structures have different conformations. The yPre-RC (**a**) is a closed complex while the hPre-RC (**b**) is an open complex.

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
