# Peer review of "The Origin Recognition Complex: From Origin Selection to Replication Licensing in Yeast and Humans"

_biology, 2023, doi:10.3390/biology13010013_

Round 1
Reviewer 1 Report
Comments and Suggestions for Authors
This review, The Origin Recognition Complex: From Origin Selection to Replication Licensing in Yeast and Humans” by Bik-Kwoon Tye and Yuanliang Zhai is a masterpiece. It explains the structure and action of the eukaryotic ORC on DNA in a way that is very easy to understand, without getting caught up in minute details, thereby making it highly accessible to a wide audience. Comparison of the yeast ORC to the human ORC is especially nice, as they have some important differences, but they are more similar than they are dissimilar. The possible reasons for the differences in ORC behavior in yeast vs humans follows a beautiful logic flow that makes complete sense, and lifts the veil off of a long held mystery. It is refreshing to see the two initiators compared side-by-side and explained so clearly. This is a lovely piece of work and I feel it should be quickly published essentially as is. The only possible small suggestions are below.
Suggestions:
1) For the structures in figure 6, might be good to have their identifier for retrieval from the structural database provided either within the figure or in the legend. This way, the reader can easily retrieve the structures for further examination.
2) in Fig. 4, the panels are labelled (i), (ii) and (iii). Yet the entire panel is labelled (b), as if it were initially created as a second panel of a larger figure. The authors might consider either removing the (b), and possibly labelling the three sets of panels as a), b, and c) (or to leave as (i), (ii) and (iii)).
3) A few grammatical changes
Line 29: “…replisome at a replication origin…”
Line 31: “binds a replication origin…”
Line 42/43: “… in segments referred to as replicons, …”
Line 48: The first instance, MGS should be placed in parenthesis just after Meier-Gorlin Syndrome is spelled out.
Line 60: subunits should not be plural: i.e. should read “…is a six subunit complex…”
Author Response
Reviewer 1:
This review, The Origin Recognition Complex: From Origin Selection to Replication Licensing in Yeast and Humans” by Bik-Kwoon Tye and Yuanliang Zhai is a masterpiece. It explains the structure and action of the eukaryotic ORC on DNA in a way that is very easy to understand, without getting caught up in minute details, thereby making it highly accessible to a wide audience. Comparison of the yeast ORC to the human ORC is especially nice, as they have some important differences, but they are more similar than they are dissimilar. The possible reasons for the differences in ORC behavior in yeast vs humans follows a beautiful logic flow that makes complete sense, and lifts the veil off of a long held mystery. It is refreshing to see the two initiators compared side-by-side and explained so clearly. This is a lovely piece of work and I feel it should be quickly published essentially as is. The only possible small suggestions are below.
We thank the reviewer for his/her gracious comments.
Suggestions:
1) For the structures in figure 6, might be good to have their identifier for retrieval from the structural database provided either within the figure or in the legend. This way, the reader can easily retrieve the structures for further examination.
Identifiers have been added.
2) in Fig. 4, the panels are labelled (i), (ii) and (iii). Yet the entire panel is labelled (b), as if it were initially created as a second panel of a larger figure. The authors might consider either removing the (b), and possibly labelling the three sets of panels as a), b, and c) (or to leave as (i), (ii) and (iii)).
We have removed the (b) in figure 4.
3) A few grammatical changes
Line 29: “…replisome at a replication origin…”
done
Line 31: “binds a replication origin…”
done
Line 42/43: “… in segments referred to as replicons, …”
done
Line 48: The first instance, MGS should be placed in parenthesis just after Meier-Gorlin Syndrome is spelled out.
done
Line 60: subunits should not be plural: i.e. should read “…is a six subunit complex…”
done

Reviewer 2 Report
Comments and Suggestions for Authors
This is a clever set of experiments based on a thoughtful approach that came from combining the cryo-EM structures of ORC DNA binding of both yeast and metazoans with protein sequence alignment to create a yeast ORC4p reminiscent of metazoan ORC4p. The use genome-wide approaches (including ChIP-seq and BrdU-IP-seq) to show this to be successful in shifting yeast ORC to a binding mode more reminiscent of metazoan, with apparently much less sequence specificity and binding more based on chromatin structure. This study postulates, and then largely answers the decades-old question of how yeast ORC shows sequence-specific DNA binding while metazoan ORC does not.
Line 20 – the word “avoid” can be interpreted as very strong. Such conflicts naturally occur and are dealt with, albeit sometimes with some difficulty. Perhaps a word such as “limit” or “manage” might more accurately reflect the situation.
Line 21 – calling yeast a “perfect” model is too strong. “Excellent” or even “outstanding” would be better choices.
Line 38 – Since origin licensing is being discussed, should not “and MCMs” (and an appropriate reference) be added to the word “ORC” here? Since the CDK phosphorylation of both play critical roles in licensing.
Line 96 – “superposed” should be “superimposed” I believe.
Lines 101 and 102 – I believe the authors should be referencing Figs 1e and 1f, respectively, in these lines.
Line 128 – I believe this sentence is meant to say “…are indistinguishable from the three DNA…” Otherwise the meaning of the sentence is very unclear.
Line 143 – saying that ORC binding with this mutation “is no longer base sequence specific” is far too strong a statement based on the experiments carried out. The complex likely has a dramatically decreased sequence specificity; but as DNA binding and sequence specificity was not evaluated biochemically, the sentence should be more like “shows an apparent decrease in sequence specificity”. As a result the “Instead,” at the initiation of the subsequent sentence should be eliminated.
Line 220 – eliminate “, which the budding yeast accomplishes so well.” The phrase is gratuitous.
Line 229 – this sentence should be clarified.
Line 234 – should be clarified. Until now the authors have spoken entirely of human/metazoan replication showing complete lack of sequence specificity. If they are now saying that it is sequence specific in a minority of cases, it needs to be explained how this is the case.
Author Response
Reviewer 2:
This is a clever set of experiments based on a thoughtful approach that came from combining the cryo-EM structures of ORC DNA binding of both yeast and metazoans with protein sequence alignment to create a yeast ORC4p reminiscent of metazoan ORC4p. The use genome-wide approaches (including ChIP-seq and BrdU-IP-seq) to show this to be successful in shifting yeast ORC to a binding mode more reminiscent of metazoan, with apparently much less sequence specificity and binding more based on chromatin structure. This study postulates, and then largely answers the decades-old question of how yeast ORC shows sequence-specific DNA binding while metazoan ORC does not.
We appreciate the reviewer‘s encouraging comments.
Line 20 – the word “avoid” can be interpreted as very strong. Such conflicts naturally occur and are dealt with, albeit sometimes with some difficulty. Perhaps a word such as “limit” or “manage” might more accurately reflect the situation.
Changed to “limit” in line 32.
Line 21 – calling yeast a “perfect” model is too strong. “Excellent” or even “outstanding” would be better choices.
Changed to “excellent” in line 33.
Line 38 – Since origin licensing is being discussed, should not “and MCMs” (and an appropriate reference) be added to the word “ORC” here? Since the CDK phosphorylation of both play critical roles in licensing.
“MCMs” added with appropriate references added in lines 50-51.
Line 96 – “superposed” should be “superimposed” I believe.
“superposed” is a better word in our context according to Oxford dictionary: Superpose: place (something) on or above something else, especially so that they coincide. Superimpose: place or lay (one thing) over another, typically so that both are still evident
Lines 101 and 102 – I believe the authors should be referencing Figs 1e and 1f, respectively, in these lines.
Figs 1e and 1f show only ORC. Not ORC over OCCM.
Line 128 – I believe this sentence is meant to say “…are indistinguishable from the three DNA…” Otherwise the meaning of the sentence is very unclear.
A-T or T-A base pairs are indistinguishable in the minor groove. Comma added after “minor groove” in line 140 and references added in line 141 to clarify this point.
Line 143 – saying that ORC binding with this mutation “is no longer base sequence specific” is far too strong a statement based on the experiments carried out. The complex likely has a dramatically decreased sequence specificity; but as DNA binding and sequence specificity was not evaluated biochemically, the sentence should be more like “shows an apparent decrease in sequence specificity”. As a result the “Instead,” at the initiation of the subsequent sentence should be eliminated.
Changes made as suggested in line 156-157. Thank you.
Line 220 – eliminate “, which the budding yeast accomplishes so well.” The phrase is gratuitous.
Eliminated in line 236 as suggested.
Line 229 – this sentence should be clarified.
Clarification added in line 243
Line 234 – should be clarified. Until now the authors have spoken entirely of human/metazoan replication showing complete lack of sequence specificity. If they are now saying that it is sequence specific in a minority of cases, it needs to be explained how this is the case.
We are referring to the alternative mechanisms such as PTM of histones used in humans for site-specific recruitment of ORC, not sequence specificity.